# Origin of competing charge density waves in kagome metal ScV$_6$Sn$_6$

Kang Wang [1], Siyu Chen [1,2], Sun-Woo Kim [1] ✉ & Bartomeu Monserrat [1,2] ✉

Understanding competing charge density wave (CDW) orders in the bilayer kagome metal ScV$_6$Sn$_6$ remains challenging. Experimentally, upon cooling, short-range order with wave vector $\mathbf{q}_2 = (\frac{1}{3}, \frac{1}{3}, \frac{1}{2})$ forms, which is subsequently suppressed by the condensation of long-range $\mathbf{q}_3 = (\frac{1}{3}, \frac{1}{3}, \frac{1}{3})$ CDW order at lower temperature. Theoretically, however, the $\mathbf{q}_2$ CDW is predicted as the ground state, leaving the CDW mechanism elusive. Here, using anharmonic phonon-phonon calculations combined with density functional theory, we predict a temperature-driven structural phase transitions from the high-temperature pristine phase to the $\mathbf{q}_2$ CDW, followed by the low-temperature $\mathbf{q}_3$ CDW, explaining experimental observations. We demonstrate that semi-core electron states stabilize the $\mathbf{q}_3$ CDW over the $\mathbf{q}_2$ CDW. Furthermore, we find that the out-of-plane lattice parameter controls the competing CDWs, motivating us to propose compressive bi-axial strain as an experimental protocol to stabilize the $\mathbf{q}_2$ CDW. Finally, we suggest Ge or Pb doping at the Sn site as another potential avenue to control CDW instabilities. Our work provides a full theory of CDWs in ScV$_6$Sn$_6$, rationalizing experimental observations and resolving earlier discrepancies between theory and experiment.

Kagome materials have emerged as promising platforms to study novel quantum phases of matter that arise from the interplay between lattice geometry, band topology, and electronic correlations[1–3]. Typical electronic band structures of kagome materials include Dirac points, flat bands, and van Hove singularities, serving as rich sources for a variety of structural and electronic instabilities[4–8]. Indeed, exotic electronic states such as superconductivity[9–11], charge density waves (CDWs)[9,12–15], pair density waves[16], non-trivial topological states[9,17], and electronic nematicity[18], have been observed in representative kagome metals $A$V$_3$Sb$_5$ ($A$=K, Rb, and Cs)[19]. Of these, the CDW state exhibits unconventional properties including time-reversal and rotational symmetry breaking[12,18,20–26], and an unconventional interplay with superconductivity featuring a double superconducting dome[27–30]. This has sparked remarkable interest and controversy, prompting the exploration of other material candidates in the quest for a comprehensive understanding of the CDW state in kagome materials.

In this context, the newly discovered bilayer kagome metals $R$V$_6$Sn$_6$ ($R$ is a rare-earth element) have attracted much attention[31–51].

Among the $R$V$_6$Sn$_6$ series, the non-magnetic ScV$_6$Sn$_6$ compound is the only one that has been reported to undergo a CDW transition, which occurs below $T_{\text{CDW}} \approx 92$ K[38]. The wave vector $\mathbf{q}_3 = (\frac{1}{3}, \frac{1}{3}, \frac{1}{3})$, often described as corresponding to a $\sqrt{3} \times \sqrt{3} \times 3$ periodicity in real space, has been identified as the ordering vector of the CDW state through x-ray diffraction (XRD)[38], neutron diffraction[38], and inelastic x-ray scattering (IXS)[39,40]. The in-plane $\sqrt{3} \times \sqrt{3}$ periodicity has been further confirmed using surface-sensitive techniques such as scanning tunneling microscopy[41,42] and angle-resolved photoemission spectroscopy[42,43]. It has also been established that the CDW distortion is dominated by out-of-plane displacements of Sc and Sn atoms[38,47], which exhibit strong electron-phonon coupling[39,40,42,48]. Interestingly, short-range order with $\mathbf{q}_2 = (\frac{1}{3}, \frac{1}{3}, \frac{1}{2})$, corresponding to $\sqrt{3} \times \sqrt{3} \times 2$ periodicity, has been detected above $T_{\text{CDW}}$, but it is supressed below $T_{\text{CDW}}$ when the dominant $\mathbf{q}_3$ CDW develops[39,40,48].

On the theoretical front, density functional theory (DFT) calculations have reproduced the competing CDW orders[47], but all previous DFT studies have found that the $\mathbf{q}_2$ CDW order is the lowest energy

[1]Department of Materials Science and Metallurgy, University of Cambridge, Cambridge, UK. [2]Cavendish Laboratory, University of Cambridge, Cambridge, UK. ✉e-mail: swk38@cam.ac.uk; bm418@cam.ac.uk

ground state[39,47,52–54], in stark contrast to experimental reports. Various mechanisms have been proposed to explain the observed $\mathbf{q}_3$ CDW ground state, including configurational entropy[52], the order-by-disorder mechanism[53], and large fluctuations from flat phonon soft modes[54]. These scenarios are based on harmonic phonon dispersions, which exhibit multiple imaginary phonon modes and a theory capable of quantitatively explaining the temperature-driven phase transition from the $\mathbf{q}_2$ order to the $\mathbf{q}_3$ CDW order is missing. Moreover, the inclusion of electronic temperature in DFT calculations is also insufficient to predict the observed CDW transition, which is overestimated by thousands of degrees. These discrepancies between theoretical models and experimental observations pose a key challenge to achieving a comprehensive understanding of CDW states in bilayer kagome metals.

In this work, we present a first principles theory of CDW order in ScV$_6$Sn$_6$ that explains the reported experimental observations and resolves earlier discrepancies between theory and experiment. Our theory is based on the inclusion of anharmonic phonon-phonon interactions, and captures the observed temperature dependence of charge orders, with a $\mathbf{q}_2$ distortion occurring at a higher temperature that is subsequently suppressed by the dominant low-temperature $\mathbf{q}_3$ charge ordering. Moreover, we calculate a phase diagram comparing the two charge orders as a function of the in-plane and out-of-plane lattice parameters, and suggest a clear pathway for using compressive bi-axial strain to experimentally stabilize a CDW with a dominant $\mathbf{q}_2$ wave vector in ScV$_6$Sn$_6$. We also predict that Ge or Pb doping at the Sn site can control CDW order, and propose ScV$_6$Pb$_6$ as another CDW material within the 166 kagome family. Finally, we rationalize discrepant earlier theoretical models by highlighting that the relative stability of the competing CDW states in ScV$_6$Sn$_6$ exhibits a complex dependence on the number of valence electrons included in the calculations and on the out-of-plane lattice parameter.

## Results and discussion

The high temperature pristine phase of ScV$_6$Sn$_6$ crystallizes in the hexagonal space group $P6/mmm$ (Fig. 1a). The primitive cell contains one Sc atom (Wyckoff position $1a$), six equivalent V atoms ($6i$), and three pairs of nonequivalent Sn atoms, labeled as Sn1 ($2e$), Sn2 ($2d$), and Sn3 ($2c$). The crystal structure consists of two kagome layers of V and Sn1 atoms, a triangular layer of Sc and Sn3 atoms, and a honeycomb layer of Sn2 atoms. In the two kagome layers, the Sn1 atoms buckle in opposite directions relative to each V kagome sublattice. The Sn1 and Sc atoms form a chain along the c axis. Notably, the smallest $R$-site ion radius of ScV$_6$Sn$_6$ among the $R$V$_6$Sn$_6$ series leads to the formation of

Sn1-Sc-Sn1 trimers mediated by a shortening of the Sc-Sn1 bonds and an elongation of the Sn1-Sn1 bonds along the chain. This unique structural feature results in more space for the Sn1-Sc-Sn1 trimers to vibrate along the c direction, a characteristic absent in other $R$V$_6$Sn$_6$ compounds without a trimer formation, which has been demonstrated to be crucial to the formation of the CDW[44,48]. Motivated by this observation, the calculations reported below are performed using the PBEsol exchange-correlation functional[55], which gives better agreement with the experimentally measured out-of-plane lattice parameter compared to the often-used PBE exchange-correlation functional[56]. Nonetheless, the overall conclusions of our work are independent of the exchange-correlation functional used (See Supplementary Note 2).

Figure 1d shows the calculated harmonic and anharmonic phonon dispersions of pristine ScV$_6$Sn$_6$. The harmonic phonon dispersion shows multiple dynamical instabilities, which appear as imaginary frequencies in the phonon dispersion (represented by the gray area in Fig. 1d). The harmonic instabilities span a wide region of the Brillouin zone, including the whole $q_z = \frac{1}{2}$ plane represented by the $A$-$L$-$H$-$A$ closed path, and also including other values of $q_z$, for example the $K'$ point at the $q_z = \frac{1}{3}$ plane. Specifically, the calculated harmonic instabilities include the CDW instabilities reported experimentally with wave vectors $\mathbf{q}_3$ and $\mathbf{q}_2$, which correspond to the $K'$ and $H$ points of the Brillouin zone, respectively; but also include many other instabilities. Interestingly, when anharmonic phonon-phonon interactions[57–59] are included, most harmonic instabilities disappear, leaving only two at the $K'$ and $H$ points at finite temperatures. This aligns with available experimental reports, which observe only those two instabilities.

Figure 1b, c illustrate the CDW structures optimized along the imaginary phonon modes at $\mathbf{q}_3$ and $\mathbf{q}_2$, respectively. The distortions of both CDW orders mainly occur along the Sn1-Sc chains, where the Sn1-Sc-Sn1 trimers are displaced to form $\times 3$ and $\times 2$ CDW periodicities along the c axis for the $\mathbf{q}_3$ and $\mathbf{q}_2$ orders, respectively. The $\mathbf{q}_3$ CDW order involves three trimers exhibiting a stationary-up-down pattern while the $\mathbf{q}_2$ CDW order involves two trimers with an up-down pattern. In both CDW orders, the trimers in one Sc1-Sn chain alternate their displacement along the c axis relative to the other Sc1-Sn chains, resulting in the $\sqrt{3} \times \sqrt{3}$ in-plane periodicity.

To further characterize the CDW of ScV$_6$Sn$_6$, we investigate the temperature dependence of phonon dispersions using anharmonic calculations within the stochastic self-consistent harmonic approximation[57–59]. Figure 2a displays the anharmonic phonon dispersion of pristine ScV$_6$Sn$_6$ at temperatures of 0 K, 50 K, 100 K and 200 K. The majority of phonon branches show a negligible temperature dependence, with the key exception of a pronounced

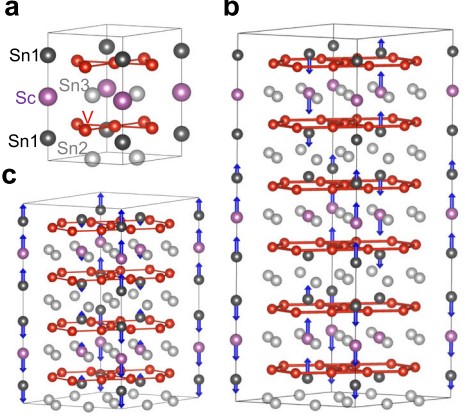

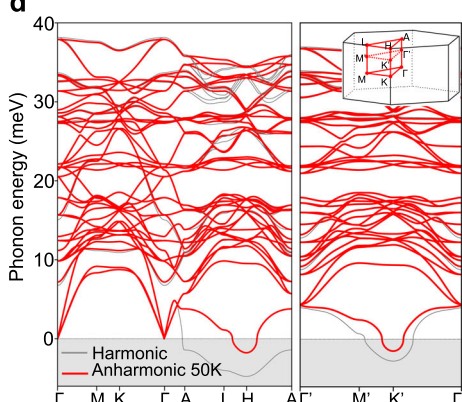

**Fig. 1 | CDW instabilities in pristine ScV$_6$Sn$_6$ and corresponding CDW structures. a** Crystal structure of pristine ScV$_6$Sn$_6$. **b, c** CDW displacement patterns (blue arrows) for the **b** $\mathbf{q}_3$ ($\sqrt{3} \times \sqrt{3} \times 3$) and **c** $\mathbf{q}_2$ ($\sqrt{3} \times \sqrt{3} \times 2$) orders. The arrows are calculated through the structural difference of the relaxed CDW structure and the pristine structure. **d** Harmonic (gray) and anharmonic (red) phonon dispersions of pristine ScV$_6$Sn$_6$. The inset shows the Brillouin zone of pristine ScV$_6$Sn$_6$ with certain high symmetry points labeled. The prime notation designates the high symmetry points on the $q_z = \frac{1}{3}$ plane.

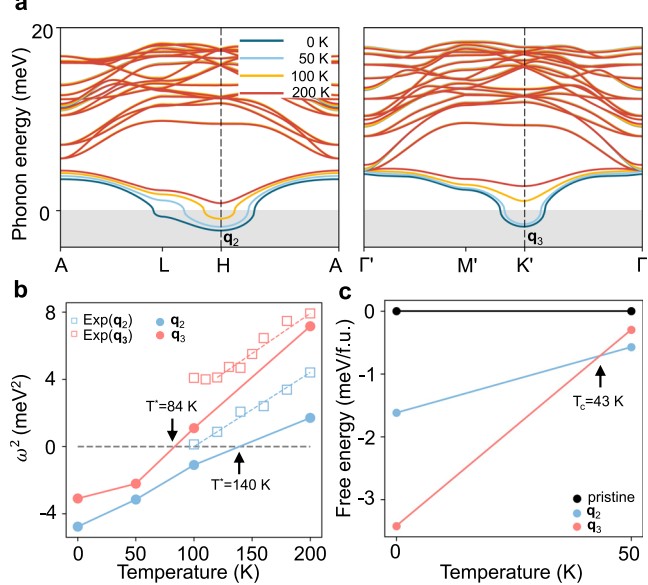

**Fig. 2 | Anharmonic phonon dispersions and anharmonic free energy.**
**a** Anharmonic phonon dispersions at 0 K, 50 K, 100 K, and 200 K. **b** Squared phonon frequency $\omega^2$ of the lowest energy phonon modes at the H and $K'$ points with respect to ionic temperature. The calculated anharmonic frequencies are shown in circles and the experimental results[40] are represented by squares. **c** Helmholtz free energy of the two CDW structures compared to the pristine structure.

softening of the lowest energy phonon branch at $\mathbf{q}_2$ (H point) and $\mathbf{q}_3$ ($K'$ point) upon cooling. The squared frequency $\omega^2$ of the soft modes should exhibit a linear behavior with respect to temperature near the vanishing point[60], and we use a linear fit to extract the transition temperatures $T^*$ associated with each phonon softening (Fig. 2b). At high temperature (200 K), the pristine ScV₆Sn₆ phase is dynamically stable and the phonon frequencies at both $\mathbf{q}$ vectors are real. Interestingly, the frequency at $\mathbf{q}_2$ becomes imaginary at a higher temperature of 140 K compared to the temperature of 84 K at which the frequency at $\mathbf{q}_3$ becomes imaginary. This implies that, starting from the high temperature pristine phase, the first distortion to develop with decreasing temperature is that associated with $\mathbf{q}_2$, rationalizing the short-range $\mathbf{q}_2$ order above the $\mathbf{q}_3$ CDW transition temperature observed in XRD[48] and IXS[40].

Our work clearly identifies anharmonic phonon-phonon interaction as the driving mechanism underlying the observed temperature-induced CDW transition in ScV₆Sn₆. In this context, our anharmonic phonon results exhibit remarkable quantitative agreement with experimental data (Fig. 2b). This should be contrasted with the significant overestimation of temperature scales (with critical temperatures of 2000 K and 5500 K) obtained by adjusting electronic temperature only via changing smearing values in harmonic phonon calculations (See Supplementary Note 3).

Comparing the calculated Helmholtz free energy between the $\mathbf{q}_2$ and $\mathbf{q}_3$ CDW orders, we observe a crossover from $\mathbf{q}_2$ to $\mathbf{q}_3$ CDWs (Fig. 2c). The $\mathbf{q}_2$ CDW order is more stable at high temperature and the free energy difference decreases as the temperature decreases. The crossover occurs approximately at $T_c = 43$ K, which is lower than the onset of the $\mathbf{q}_3$ CDW instability $T^* = 84\,K$ in Fig. 2b. The predicted transition temperature $T_c$ of the $\mathbf{q}_3$ CDW ($\approx 43$ K) is lower than experimental values of 92 K[61] and 84 K[48]. This quantitative discrepancy is attributed to the sensitivity of $T_c$ on the volume of the system, as discussed in detail below, and more generally to the inherent temperature-independent limitations of DFT calculations, such as those arising from the choice of exchange-correlation functional. At 0

K, the $\mathbf{q}_3$ CDW structure is more stable by 1.80 meV/f.u. compared to the $\mathbf{q}_2$ CDW structure.

Our DFT calculations correctly predict the $\mathbf{q}_3$ CDW to be the lowest energy ground state, consistent with experiments[38–40,48]. Puzzlingly, all earlier DFT studies[47,52,53] had predicted the $\mathbf{q}_2$ distortion to be the ground state, in stark contrast to the experimental reports and to our results. To rationalize this, we highlight that in our calculations we have established that the inclusion of electronic semi-core states in the valence is necessary to stabilize the $\mathbf{q}_3$ CDW order (Fig. 3). Using a standard pseudopotential with valence electrons $3d^14s^2$ for Sc atoms and $5s^25p^2$ for Sn atoms, which excludes the semi-core states from the valence, the total electronic energy of the $\mathbf{q}_2$ CDW is lower than that of the $\mathbf{q}_3$ CDW by 0.52 meV/f.u. (Fig. 3a), a conclusion reached by previous DFT calculations[47,52–54]. The $\mathbf{q}_2$ CDW remains the ground state in the free energy, stabilized by 0.44 meV/f.u. over the $\mathbf{q}_3$ CDW (Fig. 3b). The free energy includes both total electronic energy and phonon energy, with phonon contributions encompassing both harmonic and anharmonic zero-point energy at 0 K. By contrast, a pseudopotential that includes the $s$ and $p$ semi-core states in Sc atoms and the $d$ semi-core states in Sn atoms as valence states, predicts the $\mathbf{q}_3$ CDW to be the ground state, regardless of phonon contributions. The $\mathbf{q}_3$ CDW structure is further stabilized upon consideringng phonon contributions, increasing the energy difference between the $\mathbf{q}_2$ and $\mathbf{q}_3$ CDWs from 0.47 to 1.80 meV/f.u. (Fig. 3a, b).

To gain further insight into the role of the pseudopotential, we analyse the effect of semi-core states on the electronic structures of the $\mathbf{q}_2$ and $\mathbf{q}_3$ distorted phases (Fig. 3c, d). We find that including semi-core states shifts the overall band dispersions and DOS of the occupied states upward, leading to a reduced total energy gain of both CDW states upon formation from the pristine structure (Fig. 3a). Specifically, the total energy gain of the CDW structures over the pristine structure decreases from −3.43 to −1.95 meV/f.u. for the $\mathbf{q}_2$ CDW, and from −2.91 to −2.42 meV/f.u. for the $\mathbf{q}_3$ CDW. The larger reduction in total energy gain for the $\mathbf{q}_2$ CDW is attributed to more significant changes in its electronic structure arising from larger structural changes in the $\mathbf{q}_2$ CDW, particularly in the bond lengths in the Sn1-Sc-Sn1 chains. These bond lengths change by up to 0.065 Å in the $\mathbf{q}_2$ CDW, compared to a maximum change of 0.022 Å in the $\mathbf{q}_3$ CDW upon the inclusion of semi-core states (see Supplementary Table S4 for details). This demonstrates that semi-core effects have a more pronounced impact on the $\mathbf{q}_2$ CDW than on the $\mathbf{q}_3$ CDW. Furthermore, unlike in the $\mathbf{q}_3$ CDW state, we observe that semi-core states affect the electronic structure near the Fermi level in the $\mathbf{q}_2$ CDW. The atom-projected DOS shows that the Sc and Sn1 states, which are responsible for the CDW state, are particularly influenced, suggesting that the corrections in the Sn1-Sc-Sn1 chains alter the Fermi surface and associated properties.

The validity of our theoretical calculations is confirmed by cross-checking with pseudopotential-free all-electron WIEN2K calculations (see Supplementary Note 4). We find that the VASP results, obtained by including semi-core states as valence, show remarkable agreement with the WIEN2K results in terms of the total energy of both pristine and CDW structures, as well as their atomic and electronic structures. Overall, this demonstrates that the inclusion of semi-core electron states in the valence is necessary to obtain a theoretical model that correctly predicts the ground state CDW order of ScV₆Sn₆. This explains and resolves the outstanding discrepancy between theory and experiment, and provides a foundation for the predictive model of the CDW state in bilayer kagome ScV₆Sn₆ described above. Additionally, we note that the calculated energy difference between the two CDWs is small (less than 2 meV/f.u.), highlighting the competing nature of the two CDWs in ScV₆Sn₆. This suggests that the competition between the two CDW orders can be easily manipulated via external perturbations, as discussed in detail below.

Having established the correct theory of the competing CDWs in ScV₆Sn₆, we explore the phase diagram in lattice parameter space

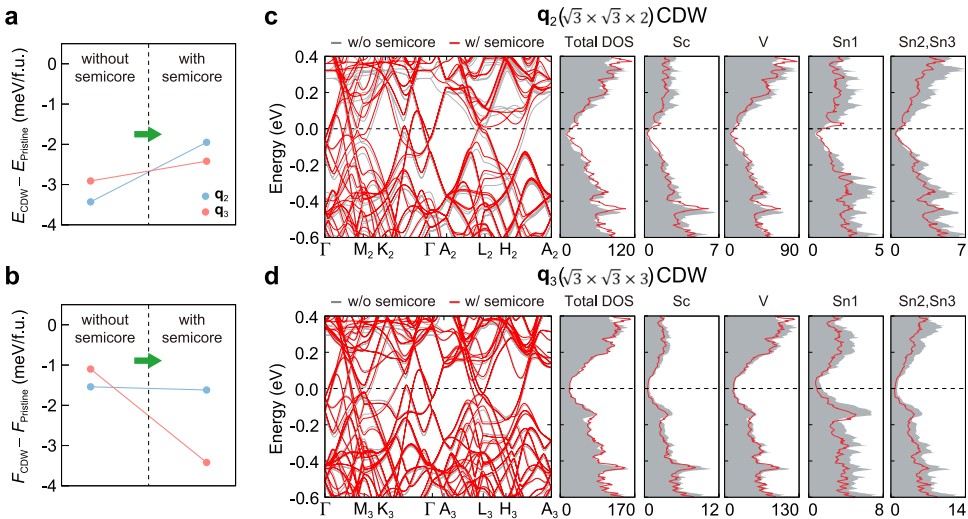

**Fig. 3 | Effect of semi-core states and anharmonic quantum effects on the energetics between CDW states. a** Total electronic energy and (**b**) free energy of CDW states relative to the pristine state at 0 K with and without including semi-core states as valence states. The free energy contains total electronic energy and phonon energy that arises from both harmonic and anharmonic quantum effects. **c**, **d** Electronic band structures and density of states (DOS) calculated with and without treating semi-core states as valence states for (**c**) the $q_2$ and **d** the $q_3$ CDW states. The semi-core states refer to the $3s$ and $3p$ orbitals of Sc atoms and the $4d$ orbitals of Sn atoms.

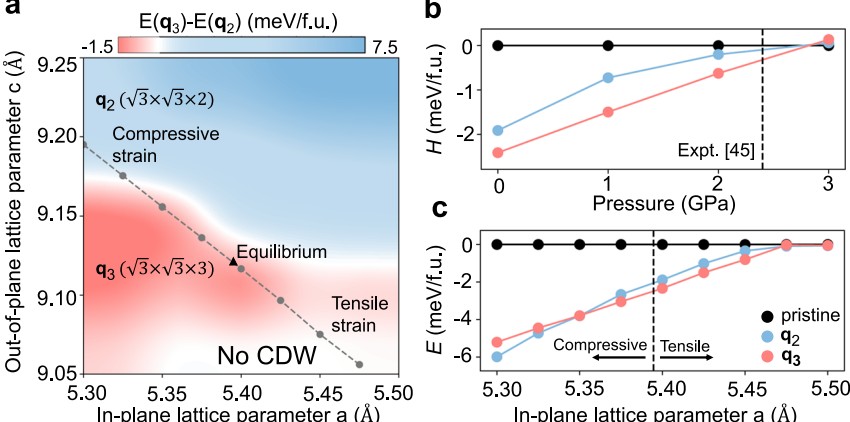

**Fig. 4 | Manipulation of competing CDWs in ScV$_6$Sn$_6$ via strain and pressure. a** Phase diagram of the competing $q_2$ and $q_3$ CDW orders in the in-plane and out-of-plane lattice parameter space. Color bar represents the total energy difference between the $q_2$ and $q_3$ structures. The gray dashed line shows the change of lattice parameters under in-plane biaxial strain. **b** Relative enthalpy $H = E + PV$ of the $q_2$ and $q_3$ CDWs to pristine structure. Vertical dashed line indicates the experimental value of critical pressure of 2.4 GPa that CDW orders are totally suppressed[45]. **c** Total energy of the $q_2$ and $q_3$ CDWs as a function of in-plane biaxial strain.

which provides important insights for manipulating the competing CDWs. Figure 4a shows a phase diagram of the $q_2$ and the $q_3$ CDW orders as a function of the in-plane and out-of-plane lattice parameters. We find that the out-of-plane lattice parameter holds the key to control competing the CDW orders. This can be rationalized by noting that the out-of-plane lattice parameter determines the space available for the distortions of adjacent Sn1-Sc-Sn1 trimers and thus dominates the energetics between competing CDWs. A smaller out-of-plane lattice parameter restricts the available space for Sn1-Sc-Sn1 trimers to distort, favoring an additional stationary configuration of trimers in $q_3$ CDW. Conversely, a larger out-of-plane lattice parameter facilitates the collective distortions of trimers and favors the $q_2$ CDW.

In addition, we discuss the effects of the hydrostatic pressure and in-plane biaxial strain on the competing CDWs. Hydrostatic pressure suppresses both CDW orders (Fig. 4b), as increasing pressure decreases the out-of-plane lattice parameter and thus the space available for Sn1-Sc-Sn1 trimers to distort. Our theoretical value for the critical pressure (≈ 2.8 GPa) at which the $q_3$ CDW disappears is in

remarkable agreement with the experimental value of 2.4 GPa[45]. Similarly, in-plane biaxial tensile strain leads to a decrease in the out-of-plane lattice parameter, suppressing both CDW orders (Fig. 4c). By contrast, in-plane compressive strain increases the out-of-plane lattice parameter, enhancing the stability of both CDW orders. Interestingly, the $q_2$ CDW order becomes the ground state under the compressive strain. The predicted value of critical compressive strain is just around 1%, which should be experimentally accessible. These predictions warrant further experimental studies to corroborate our findings.

Finally, we propose the doping of Pb or Ge at the Sn site as a further strategy to control competing CDWs in ScV$_6$Sn$_6$ and to explore the microscopic mechanisms underlying CDW instabilities within the 166 bilayer kagome family. To investigate the effect of doping, we fully substitute the Sn atom with other Group 14 elements, Ge and Pb. Figure 5a displays the harmonic phonon dispersion of ScV$_6$Pb$_6$ (red lines) and ScV$_6$Ge$_6$ (black lines). The phonon dispersion of ScV$_6$Pb$_6$ shows imaginary modes at the $L$ and $H$ points, indicating CDW instabilities. The calculated total energy (Fig. 5b) confirms that ScV$_6$Pb$_6$ has a CDW ground state, with the $L$ distortion slightly favored over the

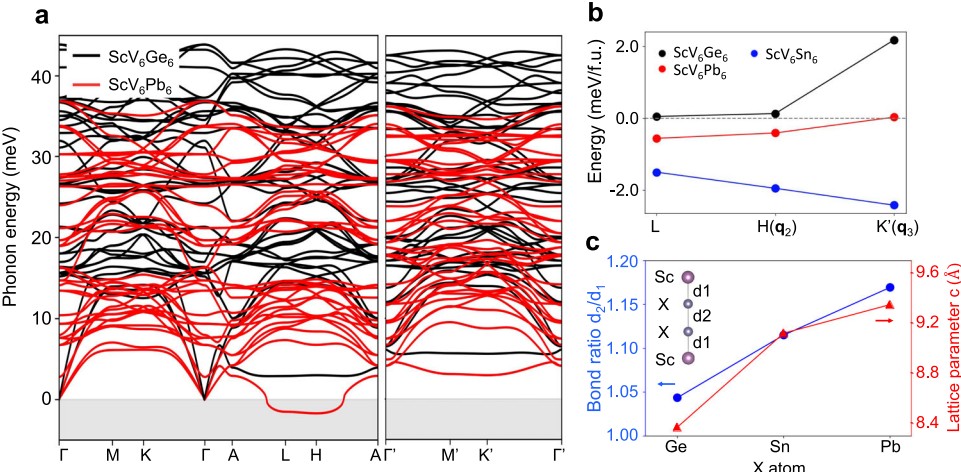

**Fig. 5 | Control of CDW instabilities in ScV$_6$Sn$_6$ via doping. a** Harmonic phonon dispersions of ScV$_6$Ge$_6$ and ScV$_6$Pb$_6$. **b** Total energy of various CDW structures relative to the pristine structure in ScV$_6$X$_6$ materials (X=Ge, Sn, and Pb). **c** Bond ratio $d_2/d_1$ in X-Sc-X chains and out-of-plane lattice parameter in the pristine structure of ScV$_6$X$_6$ materials (X=Ge, Sn, and Pb).

$\mathbf{q}_2$ distortion. We note that the $\mathbf{q}_3$ instability is not present in this compound. In contrast, ScV$_6$Ge$_6$ exhibits no imaginary modes, consistent with the calculated total energy showing that the CDW structure is unstable compared to the pristine structure. The bond ratio $d_2/d_1$ in X-Sc-X chains within the ScV$_6$X$_6$ compounds (X = Ge, Sn, and Pb) is an important indicator for the emergence of CDWs, with the ratio increasing as the atomic radius increases from Ge to Pb (Fig. 5c). A larger bond ratio is one of the necessary conditions for CDWs to emerge in this kagome system, as the CDWs displacement patterns consist of the vibration of X-Sc-X trimers. These vibrations are promoted by the larger bond difference between X-Sc and X-X atoms, which corresponds to a larger bond ratio. In the case of ScV$_6$Sn$_6$, the bond ratio is reduced to nearly 1 when Sc is substituted by larger atoms like Y or Gd, leading to the disappearance of CDWs[44]. Similarly, in ScV$_6$Ge$_6$, the bond ratio is close to 1, making trimerisation unfavorable and resulting in the absence of CDWs. Additionally, we find that there exists an optimal range of bond ratios that drive CDW formation, as the calculated energy gain of CDW states over the pristine structure diminishes in the Pb case compared to the Sn case, despite Pb having a larger bond ratio.

We note that the $\mathbf{q}_3$ CDW order is less stable compared to the pristine case in both the Ge and Pb compounds, and thus cannot be observed. In the Pb case, our theory predicts that the $L$ and $H$ ($\mathbf{q}_2$) CDWs are dominant, which we speculate is due to the larger out-of-plane lattice parameters (Fig. 5c), where we leave the detailed characterization as future work. Our results suggest that partial doping of Pb or Ge at the Sn site could suppress the $\mathbf{q}_3$ distortion, potentially leading to the dominance of the $\mathbf{q}_2$ CDW. We also predict ScV$_6$Pb$_6$ as a new CDW compound within the 166 bilayer kagome family. These predictions warrant further experimental studies to explore the role of doping in ScV$_6$Sn$_6$.

In conclusion, we have demonstrated the decisive role of temperature, captured by anharmonic phonon-phonon interactions, and volume, on the competition between the $\mathbf{q}_2$ and $\mathbf{q}_3$ charge orders in ScV$_6$Sn$_6$. Our work fully resolves the controversy between previous theoretical and experimental studies: a correct theoretical description of CDW order requires the inclusion of semi-core electron states as valence, and the use of a correct out-of-plane $c$ lattice parameter. Our theory also elucidates the experimentally observed temperature-induced CDW phase transition from a high temperature $\mathbf{q}_2$ to a low temperature $\mathbf{q}_3$ order. Finally, we predict that in-plane biaxial strain can be used to manipulate the competing CDW orders, and propose that compressive strain could be used to experimentally discover a

regime in which the $\mathbf{q}_2$ CDW dominates. We also suggest Ge or Pb doping at the Sn site can tune the competing CDW orders, and we predict ScV$_6$Pb$_6$ as a new CDW material, stimulating future experimental works. More generally, our findings contribute to the wider effort of understanding CDW states in kagome metals, including the prototypical CsV$_3$Sb$_5$[19] and magnetic FeGe[62], where the origin of the CDW states remains an open question[13,23,63–67]. Moreover, our work provides important insight into understanding and manipulating multiple CDW instabilities, such as those in CsV$_3$Sb$_5$ under doping and pressure[29,30] as well as in the recently discovered room temperature CDW kagome metal LaRu$_3$Si$_2$[68,69].

## Methods

### Electronic structure calculations

We perform density functional theory (DFT) calculations using the Vienna ab initio simulation package VASP[70,71] implementing the projector-augmented wave method[72]. We use PAW pseudopotentials with valence configurations: $3s^2 3p^6 3d^1 4s^2$ ($3d^1 4s^2$) for Sc atoms, $3s^2 3p^6 4s^2 3d^3$ ($4s^2 3d^3$) for V atoms, and $5s^2 4d^{10} 5p^2$ ($5s^2 5p^2$) for Sn atoms for the cases with (without) semi-core states. We approximate the exchange correlation functional with the generalized-gradient approximation PBEsol[55] in the calculations reported in the main text. For comparison, we also perform calculations using the PBE[56] exchange-correlation functional. We use a kinetic energy cutoff for the plane wave basis of 500 eV and a Methfessel-Paxton smearing of 0.02 eV. We use a Γ-centered **k**-point grid of size $15 \times 15 \times 8$ for the primitive cell and commensurate **k**-point grids for the supercell calculations. All the structures are optimized until the forces are below 0.005 eV/Å. We perform a cross-check of the electronic structure calculations using the CASTEP[73] package (see Supplementary Note 1.2 for details), with norm-conversing pseudopotentials generated on-the-fly (NCP19), and employing identical valence configurations as those used in VASP calculations, including semi-core states. We also use the PBEsol[55] exchange-correlation functional. We choose a kinetic energy cutoff for the plane wave basis of 1000 eV with a Gaussian smearing of 0.02 eV. We use a Monkhorst-Pack **k**-point grid with an applied half step shift if the number of **k**-points is even, which generates the exact same Gamma-centered **k**-point grid as that used in the VASP calculations. All the structures are optimized until the forces are below 0.005 eV/Å.

We also perform DFT calculations using the full-potential linearized augmented plane wave (FP-LAPW) method as implemented in the WIEN2k code[74] using the PBEsol[55] exchange-correlation functional. We

use self-consistency cycle stopping criteria of $1 \times 10^{-6}$ Ry for the energy and $1 \times 10^{-4}$ e for the charge. The radii (R) of the muffin-tin (MT) spheres are taken to be 2.40, 2.26, and 2.38 Bohr for Sc, V, and Sn atoms, respectively. $R_{MT} \times K_{max}$ is set to 8.5 (confirming that using 9 yields the same results), where $K_{max}$ is the cutoff value of the modulus of the reciprocal lattice vectors and $R_{MT}$ is the smallest MT radius. The cutoff energy separating core and valence states is set to $-136$ eV, with the valence electrons treated as $3s^2 3p^6 4s^2 3d^1$ for Sc atoms, $3s^2 3p^6 4s^2 3d^3$ for V atoms, and $4s^2 4p^6 4d^{10} 5s^2 5p^2$ for Sn atoms. All the structures are optimized until the forces are below 0.5 mRy/Bohr.

### Harmonic phonon calculations

We perform harmonic phonon calculations using the finite displacement method in conjunction with nondiagonal supercells[75,76]. The dynamical matrices are calculated on a Farey nonuniform **q** grid[77] of size $(3 \times 3 \times 2) \cup (3 \times 3 \times 3)$, which is commensurate with both $\mathbf{q}_2$ and $\mathbf{q}_3$. The final dynamical matrix is calculated through the force constant matrix on a target uniform **q** grid of size $3 \times 3 \times 6$.

### Anharmonic phonon calculations

The anharmonic phonon calculations are performed using the stochastic self-consistent harmonic approximation (SSCHA)[57–59], which is a non-perturbation method taking into account anharmonic effects at both zero and finite temperature. The free energy of the real system is variationally minimized with respect to an auxiliary harmonic system. This is done using stochastic importance sampling, in which the total energy, forces, and stresses for an ensemble of configurations of the auxiliary harmonic system are calculated using VASP. The associated electronic structure calculations are performed using a kinetic energy cutoff 300 eV, and we consider configurations commensurate with a $3 \times 3 \times 2$ supercell and a $3 \times 3 \times 3$ supercell. The number of configurations needed to converge the free energy Hessian is of the order of 1000. A Farey nonuniform **q** grid of size $(3 \times 3 \times 2) \cup (3 \times 3 \times 3)$ is used to get commensurate phonon results at both $\mathbf{q}_2$ and $\mathbf{q}_3$ (see Supplementary Note 1.1 for details). To get better prediction of the CDW transition temperature, the lattice parameters are fixed to the experimental values[38]. To obtain a more accurate energy comparison, the free energy calculations are performed using the same parameters as the electronic structure calculations.

## Data availability

The data that support the findings of this study are available within the paper and Supplementary Information. All other relevant data are available from the corresponding authors upon request.

## Code availability

The VASP code used in this study is a commercial electronic structure modeling software, available from https://www.vasp.at. The QUANTUM ESPRESSO code used in this research is open source: https://www.quantum-espresso.org/. The CASTEP code used in this study is freely available at https://www.castep.org/ for academic research. The WIEN2K code used in this study is a commercial software, available from http://susi.theochem.tuwien.ac.at/.

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

## Acknowledgements

K.W., S.-W.K., and B.M. are supported by a UKRI Future Leaders Fellowship [MR/V023926/1], and S.C. and B.M. are supported by a EPSRC grant [EP/V062654/1]. S.C. also acknowledges financial support from the Cambridge Trust and from the Winton Program for the Physics of Sustainability. B.M. also acknowledges support from the Gianna Angelopoulos Program for Science, Technology, and Innovation, and from the Winton Program for the Physics of Sustainability. The computational resources were provided by the Cambridge Tier-2 system operated by the University of Cambridge Research Computing Service and funded by EPSRC [EP/P020259/1] and by the UK National Supercomputing Service ARCHER2, for which access was obtained via the UKCP consortium and funded by EPSRC [EP/X035891/1].

## Author contributions

S.-W.K. and B.M. conceived the study and planned and supervised the research. K.W., S.C., and S.-W.K. performed the DFT calculations, K.W. performed the SSCHA calculations. All authors contributed to the analysis. K.W., S.-W.K., and B.M. wrote the manuscript with input from all authors.

## Competing interests

The authors declare no competing interests.
