## [Transparent Peer Review file · Nature Communications]

Origin of competing charge density waves in kagome metal ScV_6Sn_6

Corresponding Author: Professor Bartomeu Monserrat

Version 0:

Reviewer comments:

Reviewer #1

(Remarks to the Author)

The submitted manuscript perform first-principles based calculations on ScV_6Sn_6 . They show that anharmonic phonon-phonon interactions are crucial in the system, and accurate implementation of DFT is vital to stabilize the experimentally observed CDW order.

The calculations, including all the data analysis, seem to be carefully performed and self-consistent; and the manuscript is well written. The conclusion is, if correct, certainly an important advance in this field. I have 2 major questions:

1. The application of non-uniform grid. In the Supplementary Note 1, the authors performed several convergence tests for phonon dispersions. These tests suggest Farey results are consistent with $3 \times 3 \times 3$ uniform grid at $q_z=1/3$, and consistent with $3 \times 3 \times 2$ uniform grid at $q_z=1/2$. It is slightly different from normal understanding of "convergence tests" that the results are consistent with infinitely large Q-grid. I can understand that using large supercell is very expensive especially for anharmonic calculations, but is it possible for them to show at harmonic level, perhaps just two specific q-points (H and K'), that the Farey grid is actually yielding converging result compared to $3 \times 3 \times 6$ uniform grid? This can also be done with DFPT using single unit cell to reduce the computation efforts. This is important due to the tiny energy difference (less than 1 meV/f.u.) between the CDW structures.

2. The inclusion of semi-core states. The authors showed that the inclusion of semi-core states in the pseudopotential is crucial to obtain the correct ground state in this compound. Although strictly speaking, the error-bar of pseudopotential method itself is several orders larger than the tiny energy difference, their conclusion is still meaningful because more accurate pseudopotentials are likely to reduce the error caused by pseudopotential method itself. However, the inclusion of semi-core states is most likely to affect its electronic structure first, which eventually leads to change in the total energy. Analysis of the effect of semi-core states on the valence electronic structure would be nice and informative.

Reviewer #2

(Remarks to the Author)

In this manuscript, the authors study competing charge density waves in Kagome metal ScV_6Sn_6 . Previous DFT studies failed to predict the correct ground state of this material. In this work, the authors use SSCHA to include quantum and anharmonic effects and predict the correct ground state to be the q_3 phase. For slightly elevated temperatures, the authors predict the material will be in the q_2 state, which is also consistent with experimental results. At sufficiently high temperatures, the material retains its pristine structure, as predicted by both this work and previous DFT studies.

The alignment between this work and experimental findings is encouraging. However, several significant issues prevent me from recommending publication of this manuscript in its current form:

1. The authors introduced the concept of "multi-phonon" effects, meaning further relaxation is needed after the pristine structure is distorted along one specific soft phonon. While technically correct, this statement is well-established in the DFT community. It is difficult to imagine anyone would not relax the structure after creating a distorted structure along one specific soft phonon. From my perspective, this concept is redundant and not useful in comparison with previous DFT calculations.

2. My understanding of the differences between this work and previous DFT studies is that (a) anharmonic and quantum effects are taken into account, and (b) pseudopotentials are different. I omitted the multi-phonon part since I don't think previous works did not further relax the structure. However, in the paragraph below Fig. 3, the authors attribute the difference between this work and previous works solely to different pseudopotentials. Did the authors include anharmonic and quantum effects in Fig. 3? If not, why were these effects not considered?

3. In DFT calculations, it is not always correct that pseudopotentials including semi-core states are more accurate. From a critical standpoint, the authors are now fitting the DFT parameters (including pseudopotentials) to experimental results in this work. More evidence is needed to justify the choice of semi-core pseudopotentials.

Reviewer #3

(Remarks to the Author)

The manuscript "Origin of Competing Charge Density Waves in Kagome Metal ScV6Sn6" investigates the intricate behavior of charge density wave (CDW) orders in the kagome metal ScV6Sn6. The authors incorporate anharmonic phonon effects in density functional theory (DFT) to elucidate the temperature-driven phase transition between two CDW orders: the high-temperature short-range q_2 CDW and the low-temperature long-range q_3 CDW. They emphasize the significant roles of multi-phonon effects and semi-core electron states in stabilizing the q_3 order. The CDW transition temperature predicted from the anharmonic phonon calculations (~42 K) is comparable to the experimental value (~90 K), and is markedly more reliable than the results from harmonic phonon calculations (several thousand K) in previous works.

However, I regret to state that the results in this manuscript may not attract broad interest or generate high scientific impact, and thus do not meet the high standards of Nature Communications. A more specialized journal might be more appropriate. The primary concern is that the CDW in ScV6Sn6 has been extensively studied, including in Refs. 39, 47, and 52-54, where the out-of-plane displacement of Sn atoms is established as the primary cause of the CDW. The multi-phonon effect investigated in this work is also addressed at a phenomenological level in Ref. 47.

The results presented, although technically robust at the DFT level, may lack reliability when effects beyond DFT (eg, electron correlation) are considered. For instance, the total energy curve in Fig. 3 compares the energy from single/multi-phonon effects and calculations with or without semi-core states. The energy differences are less than 1 meV, which is generally regarded as beyond the precision of DFT.

Lastly, a more challenging yet pertinent question is why this particular kagome material, ScV6Sn6, exhibits strong anharmonic effects, whereas other kagome materials do not. Is there a more general principle that can guide the search for other kagome materials with strong anharmonic effects and intriguing CDW or possible superconducting properties?

In summary, while the manuscript provides technically sound results, its findings do not represent a significant advancement over existing literature and may benefit from being directed to a more specialized journal.

Version 1:

Reviewer comments:

Reviewer #1

(Remarks to the Author)

The authors have satisfactorily addressed my previous comments. The quality of the revised manuscript seems to be much enhanced, and the readability is much improved.

There is just one more point I would like to raise. To be fair, the strong electron-phonon coupling effect and short-range order ($1/3, 1/3, 1/2$) was also discussed (perhaps the first) in Ref. 39. So I would suggest the authors to add Ref. 39 to the appropriate positions in the introductions.

Reviewer #2

(Remarks to the Author)

The authors have addressed all issues raised in my previous report. I now recommend its publication in Nature Communications.

Reviewer #3

(Remarks to the Author)

In the revised manuscript, the authors have addressed the majority of my previous comments and made improvements to the overall clarity and depth of the study. The newly added discussion on Ge and Pb substitution for Sn is a valuable addition that could potentially inspire further experimental investigations in bilayer kagome systems. I am inclined to recommend the manuscript for publication, contingent upon addressing the following minor points:

1. While the authors have provided an argument regarding the choice of semi-core pseudopotentials, I remain somewhat unconvinced. Although the DFT calculations are technically sound and are supported by full-electron calculations using

WIEN2K, the observed energy difference is still as small as ~ 1 meV. This raises the possibility that other factors, which may not have been fully accounted for in the present DFT framework, could be influencing the results. That said, given that self-consistent anharmonic calculations and full-electron WIEN2K calculations are regarded as the most accurate DFT method, it is understandable that there may be limitations to what can be further addressed here. Nevertheless, I suggest that the authors include a brief discussion on this small energy difference in the manuscript to provide context.

2. Regarding the density of states for the q2 CDW phase with semi-core states, there is a small peak at the Fermi level associated with contributions from Sc and Sn1 atoms. This feature is somewhat unexpected, as one would typically anticipate that a CDW phase would reduce the DOS at the Fermi level, rather than produce a peak. Could the authors elaborate on the origin of this peak? Specifically, does this feature persist in the DOS without the presence of the CDW? Additionally, could this peak be related to the total energy difference between the q2 and q3 CDW phases, given that the q3 CDW does not exhibit such peaks?

If these minor comments can be addressed, I believe the manuscript would be suitable for publication.

Summary of changes

Main text :

- We have updated Figure 2c to show the free energy difference of CDW states relative to the pristine structure, which has results in a change in the predicted transition temperature T_c by 1 K.
- We have removed all discussions related to single- and multi-phonon effects from the revised manuscript.
- We have updated Figure 3 to more effectively highlight our novel findings by presenting the effects of semi-core states on both the total electronic energy and free energy, along with corresponding discussions.
- We have also included the effects of semi-core states on the electronic structures of the CDW states in Figure 3, along with detailed discussions.
- We have mentioned the validity of our results obtained using the VASP pseudopotentials compared with pseudopotential-free all-electron WIEN2K results.
- We have added Figure 5 to show other potential CDW material candidates within the 166 bilayer kagome family, and to provide important insights into the microscopic mechanisms underlying CDW instabilities.
- We have added computational details of WIEN2K calculations in Methods section.
- We have added computational details of free energy calculations in Methods section.

Supplementary information :

- We have removed all discussions related to single- and multi-phonon effects from the Supplementary Information (Supplementary Note 2.3 and Supplementary Note 5 in the previous version).
- We have removed Supplementary Note 4 in the previous version, as it is no longer required due to the inclusion of L point CDW results in the newly added Figure 5 in the main text.

- We have compared harmonic phonon dispersions obtained using the uniform $3 \times 3 \times 6$ grid and the non-uniform Farey grid of size $(3 \times 3 \times 2) \cup (3 \times 3 \times 3)$ to demonstrate the effectiveness of the Farey grid. The results of this comparison are shown in the new Supplementary Figure S1.
- We have justified the choice of VASP pseudopotentials by comparing them with newly performed pseudopotential-free all-electron WIEN2K calculation results (with new Supplementary Figures S5-S9 and Tables S3-S4 in newly added Supplementary Note 4), in terms of:
 - Total energy landscape of the pristine structure
 - Total energy gain of CDW states compared to the pristine structure
 - Bond lengths in the Sn1-Sc-Sn1 chains of the pristine and CDW structures
 - Electronic structures

Added references

Please note that the following references have been added or updated to reflect recent publications. Citation numbers reflect the reference order in the most recent version of our manuscript:

- 69 Plokhikh, I. et al. Discovery of charge order above room-temperature in the prototypical kagome superconductor $\text{La}(\text{Ru}_{1-x}\text{Fe}_x)_3\text{Si}_2$. *Commun Phys* **7**, 1–8 (2024).
- 72 Blochl, P. E. Projector augmented-wave method. *Phys. Rev. B* **50**, 17953–17979 (1994).

Response to Reviewer 1

Reviewer 1 (R) : The submitted manuscript perform first-principles based calculations on ScV₆Sn₆. They show that anharmonic phonon-phonon interactions are crucial in the system, and accurate implementation of DFT is vital to stabilize the experimentally observed CDW order.

The calculations, including all the data analysis, seem to be carefully performed and self-consistent; and the manuscript is well written. The conclusion is, if correct, certainly an important advance in this field. I have 2 major questions.

Authors (A): We are grateful to the Reviewer for taking the time and effort to read our manuscript. We also thank the Reviewer for the positive evaluation on our work.

R: 1. The application of non-uniform grid. In the Supplementary Note 1, the authors performed several convergence tests for phonon dispersions. These tests suggest Farey results are consistent with 3x3x3 uniform grid at $q_z=1/3$, and consistent with 3x3x2 uniform grid at $q_z=1/2$. It is slightly different from normal understanding of "convergence tests" that the results are consistent with infinitely large Q-grid. I can understand that using large supercell is very expensive especially for anharmonic calculations, but is it possible for them to show at harmonic level, perhaps just two specific q-points (H and K'), that the Farey grid is actually yielding converging result compared to 3x3x6 uniform grid? This can also be done with DFPT using single unit cell to reduce the computation efforts. This is important due to the tiny energy difference (less than 1 meV/f.u.) between the CDW structures.

A: We thank the Reviewer for the comment. As suggested by the Reviewer, we have performed harmonic calculations using a $3 \times 3 \times 6$ uniform **q**-point grid for comparison with the non-uniform Farey grid results (Fig. R1). The results reported in Fig. R1a demonstrate that the non-uniform Farey grid used leads to the same results as the uniform $3 \times 3 \times 6$ grid.

However, we also take this opportunity to further clarify the power of the non-uniform Farey grid. In general, calculated phonon frequencies are exact at any **q**-point commensurate with the grid used. Therefore, the phonon frequencies at the H point $(\frac{1}{3}, \frac{1}{3}, \frac{1}{2})$ are exact for the uniform $3 \times 3 \times 2$ **q**-point grid, and the phonon frequencies at the K' point $(\frac{1}{3}, \frac{1}{3}, \frac{1}{3})$ are exact for the uniform $3 \times 3 \times 3$ grid. Using uniform grids, they are both *simultaneously* exact for the uniform $3 \times 3 \times 6$ grid. Importantly, using a non-uniform Farey grid of $(3 \times 3 \times 2) \cup (3 \times 3 \times 3)$ leads to frequencies

that are also *simultaneously* exact for both H and K' points [Fig. R1(a)].

FIG. R1. **a** Calculated harmonic phonon dispersions using \mathbf{q} -point grid sizes of a $3 \times 3 \times 6$ uniform grid (red dashed line) and a $(3 \times 3 \times 2) \cup (3 \times 3 \times 3)$ non-uniform Farey grid (black solid line). **b** Calculated harmonic phonon dispersions using \mathbf{q} -point grid sizes of $3 \times 3 \times 2$ (cyan dashed line) and $3 \times 3 \times 3$ (yellow dashed line) uniform grids as well as a $(3 \times 3 \times 2) \cup (3 \times 3 \times 3)$ non-uniform Farey grid (black solid line).

We have added the relevant discussions in Supplementary Note 1 with new Supplementary Figure S1.

R: 2. The inclusion of semi-core states. The authors showed that the inclusion of semi-core states in the pseudopotential is crucial to obtain the correct ground state in this compound. Although strictly speaking, the error-bar of pseudopotential method itself is several orders larger than the tiny energy difference, their conclusion is still meaningful because more accurate pseudopotentials are likely to reduce the error caused by pseudopotential method itself. However, the inclusion of semi-core states is most likely to affect its electronic structure first, which eventually leads to change in the total energy. Analysis of the effect of semi-core states on the valence electronic structure would be nice and informative.

A: We sincerely appreciate the Reviewer for the insightful comment, which helps clarify our findings and thus their importance. Figure R2 illustrates the effect of semi-core states on the electronic structures of the \mathbf{q}_2 and \mathbf{q}_3 distorted phases. We find that including semi-core states shifts the overall band dispersions and DOS of the occupied states upward, leading to a reduced total energy gain of both CDW states upon formation from the pristine structure (Table R1). Specific

FIG. R2. **Effect of semi-core states on the electronic structures of CDW states.** a,b Electronic band structures and density of states (DOS) calculated with and without treating semi-core states as valence states for a the \mathbf{q}_2 and b the \mathbf{q}_3 CDW states. The semi-core states refer to the 3s and 3p orbitals of Sc atoms and the 4d orbitals of Sn atoms.

TABLE R1. **Total energy of CDW states relative to the pristine structure.** In the WIEN2K calculations, the cutoff energy separating core and valence states is set to -136 eV, with the valence electrons treated as $3s^23p^64s^23d^1$ for Sc atoms, $3s^23p^64s^23d^3$ for V atoms, and $4s^24p^64d^{10}5s^25p^2$ for Sn atoms.

Unit:meV/f.u.	VASP (w/o semicore)	VASP (w/ semicore)	WIEN2K
\mathbf{q}_2 CDW	-3.43	-1.95	-1.91
\mathbf{q}_3 CDW	-2.91	-2.42	-2.69

ically, the total energy gain of the CDW structures over the pristine structure decreases from -3.43 to -1.95 meV/f.u. for the \mathbf{q}_2 CDW, and from -2.91 to -2.42 meV/f.u. for the \mathbf{q}_3 CDW. The larger reduction in total energy gain for the \mathbf{q}_2 CDW is attributed to more significant changes in its electronic structure arising from larger structural changes in the \mathbf{q}_2 CDW, particularly in the bond lengths in the Sn1-Sc-Sn1 chains. These bond lengths change by up to 0.065 Å in the \mathbf{q}_2 CDW, compared to a maximum change of 0.022 Å in the \mathbf{q}_3 CDW upon the inclusion of semi-core states (see Table R2). This demonstrates that semi-core effects have a more pronounced impact on the \mathbf{q}_2 CDW than on the \mathbf{q}_3 CDW. Furthermore, unlike in the \mathbf{q}_3 CDW state, we observe that semi-core states affect the electronic structures at the Fermi level in the \mathbf{q}_2 CDW. The atom-projected DOS shows that the Sc and Sn1 states, which are responsible for the CDW state, are particularly influenced, suggesting that the corrections in the Sn1-Sc-Sn1 chains alter the Fermi surface and associated properties.

TABLE R2. **Bond lengths (in Å) in Sn1-Sc-Sn1 chains.** The bond lengths d_i are labelled in Fig. R3.

Structure	Bond	VASP (w/o semicore)	VASP (w/ semicore)	WIEN2K	Expt. [R1]
\mathbf{q}_2 CDW	d_1	2.889	2.904	2.907	-
	d_2	3.044	3.063	3.052	-
	d_3	2.922	2.928	2.931	-
	d_4	3.608	3.543	3.545	-
\mathbf{q}_3 CDW	d_1	2.955	2.967	2.972	2.995
	d_2	3.148	3.126	3.106	3.074
	d_3	2.891	2.901	2.902	2.924
	d_4	2.917	2.925	2.927	2.949
	d_5	3.553	3.552	3.576	3.530

To further validate our conclusions, we have performed additional calculations using the pseudopotential-free all-electron WIEN2K package, and we find that the WIEN2K results closely

FIG. R3. Labelling of bonds in the Sn1-Sc-Sn1 chains of **a** the \mathbf{q}_2 and **b** the \mathbf{q}_3 CDW structures.

align with those obtained using the pseudopotential that includes semi-core states. The WIEN2K calculations predict the \mathbf{q}_3 CDW as the ground state, consistent with the VASP results with semi-core states included (Table R1). The total energy gain upon CDW formation is -1.91 and -2.69 meV/f.u. for the \mathbf{q}_2 and the \mathbf{q}_3 CDW states, respectively, with magnitudes similar to those from the VASP results including semi-core states. Moreover, the bond lengths in the Sn1-Sc-Sn1 chains calculated by WIEN2K are close to those obtained from VASP with semi-core states included (with a maximum difference of 0.024 Å), and show better agreement than those obtained without semi-core states (which have a maximum difference of 0.063 Å). When comparing the results with available experimental data for the \mathbf{q}_3 CDW structure, the VASP results with semi-core states included show better agreement with the experimental data, further validating the inclusion of semi-core states over results without them (Table R2).

In the revised manuscript, we have included a detailed analysis of the electronic structures with and without semi-core states by adding the electronic structures in the revised Figure 3. Additionally, we have added a new section as Supplementary Note 4, comparing the VASP results with the WIEN2K results.

Response to Reviewer 2

Reviewer 2 (R) : In this manuscript, the authors study competing charge density waves in Kagome metal ScV_6Sn_6 . Previous DFT studies failed to predict the correct ground state of this material. In this work, the authors use SSCHA to include quantum and anharmonic effects and predict the correct ground state to be the q3 phase. For slightly elevated temperatures, the authors predict the material will be in the q2 state, which is also consistent with experimental results. At sufficiently high temperatures, the material retains its pristine structure, as predicted by both this work and previous DFT studies.

The alignment between this work and experimental findings is encouraging. However, several significant issues prevent me from recommending publication of this manuscript in its current form.

Authors (A): We are grateful to the Reviewer for taking the time and effort to read our manuscript. We also thank the Reviewer for acknowledging the significance of our work.

R: 1. The authors introduced the concept of “multi-phonon” effects, meaning further relaxation is needed after the pristine structure is distorted along one specific soft phonon. While technically correct, this statement is well-established in the DFT community. It is difficult to imagine anyone would not relax the structure after creating a distorted structure along one specific soft phonon. From my perspective, this concept is redundant and not useful in comparison with previous DFT calculations.

A: We thank the Reviewer for the comment and agree that this concept is redundant and not particularly useful in comparison with previous DFT calculations, as it is already well-established within the DFT community. In response, we have removed all discussions related to single- and multi-phonon effects from the revised manuscript. Additionally, we have updated Figure 3 to more effectively highlight our novel findings by removing the single-phonon results and showing only the final, fully relaxed multi-phonon results.

R: 2. My understanding of the differences between this work and previous DFT studies is that (a) anharmonic and quantum effects are taken into account, and (b) pseudopotentials are different. I omitted the multi-phonon part since I don't think previous works did not further relax the structure. However, in the paragraph below Fig. 3, the authors attribute the difference between this work and previous works solely to different pseudopotentials. Did the authors include anharmonic and quantum effects in Fig. 3? If not, why were these effects not considered?

A: We appreciate the Reviewer for raising this important point. As correctly noted, the key distinction of our work compared to previous studies is the inclusion of both anharmonic and quantum effects, along with the use of a different pseudopotential that incorporates semi-core states as valence states. We acknowledge that Figure 3 in the previous manuscript did not effectively highlight these aspects, as it omitted the anharmonic and quantum effects, which should have been included. During the revision, we have updated Figure 3 to display both pseudopotential effects and anharmonic and quantum effects by presenting both total energy and free energy at zero temperature (Fig. R4).

FIG. R4. **Effect of semi-core states and anharmonic quantum effects on the energetics between CDW states.** **a** Total electronic energy of CDW states relative to the pristine state at 0 K with and without including semi-core states as valence states. **b** Free energy of CDW states relative to the pristine state at 0 K with and without including semi-core states as valence states. The free energy contains total electronic energy and phonon energy that arises from both harmonic and anharmonic quantum effects.

The free energy consists of the total electronic energy and the phonon energy containing both harmonic and anharmonic contributions. Even at 0 K, phonons contribute to the free energy through both harmonic and anharmonic zero-point energy, while at higher temperatures entropy contributions also contribute. Our results show that when phonon contributions are accounted for, the energy gain of the CDW states relative to the pristine structure decreases from -3.43 to -1.54 meV/f.u. for the q_2 CDW and from -2.91 to -1.10 meV/f.u. for the q_3 CDW when the

semi-core states are excluded. The \mathbf{q}_2 CDW remains the predicted ground state when phonon contributions are included without semicore states in the pseudopotential. By contrast, when semi-core states are incorporated into the pseudopotential, the ground state becomes the \mathbf{q}_3 CDW, regardless of phonon contributions. Notably, phonon contributions increase the energy difference between the \mathbf{q}_2 and \mathbf{q}_3 CDWs from 0.47 to 1.80 meV/f.u., emphasizing the critical role of semi-core states in stabilizing the \mathbf{q}_3 CDW in this kagome material.

In the revised manuscript, we have updated Figure 3 to clarify these findings by including the above figure and removing single-phonon results, focusing solely on fully relaxed multi-phonon results. Additionally, we have included the corresponding discussions throughout the manuscript to align with the changes in Figure 3. We believe these revisions more effectively highlight our novel findings.

R: 3. In DFT calculations, it is not always correct that pseudopotentials including semi-core states are more accurate. From a critical standpoint, the authors are now fitting the DFT parameters (including pseudopotentials) to experimental results in this work. More evidence is needed to justify the choice of semi-core pseudopotentials.

A: We thank the Reviewer for this comment, which led us to clarify the justification for our chosen pseudopotential in detail, further validating our conclusions. We agree that including semi-core states in a pseudopotential does not always guarantee accuracy, even though it is designed to do so, as deviations from experimental results can arise from other inherent errors in DFT calculations. To assess this point for this kagome system, we have performed extensive additional calculations to confirm that the pseudopotential including semi-core states yields more accurate results (detailed below). Importantly, it is essential for accurately describing the ground state and thus predicting uncharted regimes beyond current experimental observations. Without the inclusion of semi-core states, the theoretical predictions fail to achieve these.

We emphasise that our DFT parameters are not constrained by experimental data, as our conclusions do not rely on any experimental input. Both in the previous manuscript and in the newly added justification of the pseudopotential and corresponding analysis (see below), we have fully accounted for lattice relaxation effects, which align well with experimental data.

We justify our choice of the semi-core pseudopotential by comparing results with those from pseudopotential-free all-electron WIEN2K calculations. Below, we demonstrate the remarkable agreement between the VASP results using the pseudopotential with semi-core states and the WIEN2K results in terms of: (i) the total energy landscape of the pristine structure as a function of lattice parameters, (ii) the total energy gain of CDW states compared to the pristine structure, (iii) bond lengths in the Sn1-Sc-Sn1 chains of the pristine and CDW structures, and (iv) electronic structures.

- (i) The total energy landscape of the pristine structure

Figure R5 shows the total energy of the pristine structure as a function of out-of-plane and in-plane lattice parameters, using pseudopotentials with and without semi-core states, compared to all-electron WIEN2K calculations. WIEN2K is frequently used as a high-accuracy reference to verify whether the pseudopotential accurately describes a system. The data clearly demonstrate that the pseudopotential including semi-core states aligns almost perfectly with WIEN2K results,

FIG. R5. Total energy of the pristine structure calculated using pseudopotential methods with and without semi-core states compared to all-electron **WIEN2K** results as a function of **a** out-of-plane lattice parameter c and **b** in-plane lattice parameter a .

indicating its superior accuracy compared to the pseudopotential without semi-core states.

- (ii) The total energy gain of CDW states compared to the pristine structure

We find that the **WIEN2K** results closely align with those obtained using the pseudopotential that includes semi-core states. The **WIEN2K** calculations predict the \mathbf{q}_3 CDW as the ground state, which is consistent with the **VASP** results when semi-core states are included (Table R3). Crucially, without semi-core states, **VASP** fails to predict the correct ground state. Quantitatively, the total energy gain upon CDW formation is -1.91 meV/f.u. for the \mathbf{q}_2 CDW and -2.69 meV/f.u. for the \mathbf{q}_3 CDW, resulting in an energy difference of 0.78 meV/f.u. between the two CDW states. This value is close to the **VASP** result with semi-core states, which is 0.47 meV/f.u. Notably, the total energy gain for the \mathbf{q}_2 CDW (-1.91 meV/f.u.) is very close to the value obtained with the pseudopotential including semi-core states (-1.95 meV/f.u.).

TABLE R3. Total energy of CDW states relative to the pristine structure. In the **WIEN2K** calculations, the cutoff energy separating core and valence states is set to -136 eV, with the valence electrons treated as $3s^23p^64s^23d^1$ for Sc atoms, $3s^23p^64s^23d^3$ for V atoms, and $4s^24p^64d^{10}5s^25p^2$ for Sn atoms.

Unit:meV/f.u.	VASP (w/o semicore)	VASP (w/ semicore)	WIEN2K
\mathbf{q}_2 CDW	-3.43	-1.95	-1.91
\mathbf{q}_3 CDW	-2.91	-2.42	-2.69

- (iii) Bond lengths in the Sn1-Sc-Sn1 chains of the pristine and CDW structures

We first compare the bond lengths in the Sn1-Sc-Sn1 chains of the pristine structure. The pseudopotential without semi-core states yields $d_1 = 2.922 \text{ \AA}$ and $d_2 = 3.274 \text{ \AA}$ (Table R4; bond lengths d_i are labelled in Fig. R6), while the pseudopotential with semi-core states gives $d_1 = 2.933 \text{ \AA}$ and $d_2 = 3.255 \text{ \AA}$. These latter values are closer to the WIEN2K and experimental results, demonstrating the improved accuracy provided by including semi-core states.

For the CDW structures, the bond lengths obtained using the pseudopotential with semi-core states again closely match those from WIEN2K, with a maximum difference of 0.024 \AA compared to the WIEN2K results. In contrast, the bond lengths obtained without semi-core states show a larger maximum difference of 0.063 \AA compared to the WIEN2K results. When comparing the VASP results with available experimental data for the \mathbf{q}_3 CDW structure, the results with semi-core states exhibit better agreement with the experiments, further validating the importance of including semi-core states for accurate predictions.

TABLE R4. **Bond lengths (in \AA) of Sn1-Sc-Sn1 chains.** The bond lengths d_i are labelled in Fig. R6. In the WIEN2K calculations, the cutoff energy separating core and valence states was set to -136 eV , with the valence electrons treated as $3s^23p^64s^23d^1$ for Sc atoms, $3s^23p^64s^23d^3$ for V atoms, and $4s^24p^64d^{10}5s^25p^2$ for Sn atoms. We use the fully relaxed lattice parameters employing PBEsol functional.

Structure	Bond	VASP (w/o semicore)	VASP (w/ semicore)	WIEN2K	Expt. [R1]
Pristine	d_1	2.922	2.933	2.938	2.971
	d_2	3.274	3.255	3.245	3.218
\mathbf{q}_2 CDW	d_1	2.889	2.904	2.907	-
	d_2	3.044	3.063	3.052	-
	d_3	2.922	2.928	2.931	-
	d_4	3.608	3.543	3.545	-
\mathbf{q}_3 CDW	d_1	2.955	2.967	2.972	2.995
	d_2	3.148	3.126	3.106	3.074
	d_3	2.891	2.901	2.902	2.924
	d_4	2.917	2.925	2.927	2.949
	d_5	3.553	3.552	3.576	3.530

FIG. R6. Labelling of bonds in the Sn1-Sc-Sn1 chains of **a** the pristine, **b** the \mathbf{q}_2 CDW, and **c** the \mathbf{q}_3 CDW structures.

- (iv) Electronic structures

As the CDW structures change with the inclusion of semi-core states, their electronic structures also change accordingly. We note that semi-core effects have a more pronounced impact on the \mathbf{q}_2 CDW than on the \mathbf{q}_3 CDW. This is evident from the larger changes in bond lengths in the Sn1-Sc-Sn1 chains for the \mathbf{q}_2 CDW (with a maximum change of 0.065 Å) compared to those in the \mathbf{q}_3 CDW (with a maximum change of 0.022 Å), as presented in Table R4. These bond length changes result in more significant alterations in the electronic structures of the \mathbf{q}_2 CDW, leading to more visible difference in the electronic structures upon including semi-core states, as shown in Fig. R7 and Fig. R8. Again, the electronic structures obtained with semi-core states remarkably match those from WIEN2K for both CDW structures.

FIG. R7. Total density of states (DOS) and atom-projected density of states (PDOS) for the $\mathbf{q}_2 (\sqrt{3} \times \sqrt{3} \times 2)$ CDW calculated using VASP with and without semi-core states, compared with WIEN2K results.

FIG. R8. Total density of states (DOS) and atom-projected density of states (PDOS) for the $\mathbf{q}_3 (\sqrt{3} \times \sqrt{3} \times 3)$ CDW calculated using VASP with and without semi-core states, compared with WIEN2K results.

We further comment on the inaccuracy of the pseudopotential without including semi-core states by demonstrating its failure to explain the experimental data. Figure R9 shows the phase diagram of the two CDW orders in lattice parameter spaces. In contrast to results with the semi-core states included, the pseudopotential without the semi-core states fails to predict the \mathbf{q}_3 CDW order across any of the lattice parameters explored.

FIG. R9. Phase diagram of the competing \mathbf{q}_2 and \mathbf{q}_3 CDW orders in the in-plane and out-of-plane lattice parameter space, calculated using **a** without and **b** with semi-core states as valence states. Color bar represents the total energy difference between the \mathbf{q}_2 and \mathbf{q}_3 structures.

We have added above discussions about the justification of the VASP pseudopotential in the newly added Supplementary Note 4.

Response to Reviewer 3

Reviewer 3 (R) : The manuscript “Origin of Competing Charge Density Waves in Kagome Metal ScV₆Sn₆” investigates the intricate behavior of charge density wave (CDW) orders in the kagome metal ScV₆Sn₆. The authors incorporate anharmonic phonon effects in density functional theory (DFT) to elucidate the temperature-driven phase transition between two CDW orders: the high-temperature short-range q^2 CDW and the low-temperature long-range q^3 CDW. They emphasize the significant roles of multi-phonon effects and semi-core electron states in stabilizing the q^3 order. The CDW transition temperature predicted from the anharmonic phonon calculations (~ 42 K) is comparable to the experimental value (~ 90 K), and is markedly more reliable than the results from harmonic phonon calculations (several thousand K) in previous works.

A: We are grateful to the Reviewer for taking the time and effort to read our manuscript and nicely summarising our work.

R: However, I regret to state that the results in this manuscript may not attract broad interest or generate high scientific impact, and thus do not meet the high standards of *Nature Communications*. A more specialized journal might be more appropriate. The primary concern is that the CDW in ScV_6Sn_6 has been extensively studied, including in Refs. 39, 47, and 52-54, where the out-of-plane displacement of Sn atoms is established as the primary cause of the CDW. The multi-phonon effect investigated in this work is also addressed at a phenomenological level in Ref. 47.

A: We respectfully disagree with the Reviewer's view on the suitability of our work for publication in *Nature Communications*, and note that the other two Reviewers also agree with us that this work is of interest and suitable for *Nature Communications* after addressing some technical questions which we do above (see response to other Reviewers).

The Reviewer is correct that there have been extensive studies on the CDW in ScV_6Sn_6 in theory and experiment, establishing the out-of-plane displacement of Sn atoms as the CDW pattern of this kagome compound. Importantly, despite these extensive studies, we emphasize again that *all* previous first-principles studies failed to predict the CDW ground state, leaving the origin of the CDW states elusive and therefore severely limiting the understanding of CDWs in ScV_6Sn_6 .

Our work is the first to provide a full quantitative first-principles theoretical model that not only resolves the ongoing controversy between theory and experiment by explaining existing experimental data, but also predicts an uncharted regime in the phase diagram of ScV_6Sn_6 . Specifically, our work forecasts a new regime dominated by the \mathbf{q}_2 CDW, which arises from biaxial compressive strain and Pb or Ge doping on a Sn site, where the latter doping scenario is a new result revealed during the revision. These predictions are expected to motivate further experiment in this fascinating family of compounds. We thus believe that our work will attract broad interest and generate high scientific impact, meeting the high standards of *Nature Communications*.

R: The results presented, although technically robust at the DFT level, may lack reliability when effects beyond DFT (eg, electron correlation) are considered. For instance, the total energy curve in Fig. 3 compares the energy from single/multi-phonon effects and calculations with or without semi-core states. The energy differences are less than 1 meV, which is generally regarded as beyond the precision of DFT.

A: We thank the Reviewer for this comment. While we acknowledge that the calculated energy differences between the CDW states in this system are small, we would like to emphasise that the trends predicted by our comprehensive first principles calculations are robust. In addition to being the first to correctly predict the equilibrium ground state, our work provides a broader phenomenology of the competing CDW orders across a wide range of lattice parameters and pressure.

The reliability of our theoretical calculations is demonstrated by cross-checking with high-accuracy reference calculations at the DFT level, performed using the pseudopotential-free all-electron method with the WIEN2K code. WIEN2K is widely regarded as a benchmark for DFT calculations due to its high precision [R2], and a detailed comparison of our results with WIEN2K calculations is provided in our response to the 2nd Reviewer's comments above.

Furthermore, the conclusions of our findings have been validated by considering advanced methods beyond conventional DFT, which we discuss in detail below. Moreover, we highlight that our theoretical predictions, made without any experimental input, show strong alignment with existing experimental data and further predict new phenomena that await experimental verification.

We now address the Reviewer's concern regarding the reliability of our conclusions, particularly when considering effects beyond conventional DFT. In our study, we have considered effects beyond standard DFT, such as free energy contributions at finite temperatures and electron correlations. The most significant contribution to the free energy in this system comes from phonons, as these have been observed to soften with temperature, driving the CDWs in this compound. To account for these effects, we employ state-of-the-art methods that incorporate anharmonic phonon contributions to the free energy. Although anharmonic phonon effects are typically more pronounced at finite temperatures, they also contribute at 0 K through zero-point energy. Importantly, these anharmonic effects can still influence the stability and energetics of CDW states. As shown in Fig. R10, our DFT total energy results remain robust even when including free energy contributions from both harmonic and anharmonic zero-point energy. Notably, only when semi-core states are included does the \mathbf{q}_3 CDW become the ground state.

FIG. R10. Total electronic energy and free energy at 0 K of CDW states relative to the pristine structure for **a** without and **b** with including semi-core states as valence states. The free energy contains total electronic energy and phonon energy that arises from both harmonic and anharmonic quantum effects.

We also address correlation effects by performing DFT+ U calculations. Since the CDW pattern corresponds to out-of-plane displacements in the Sc-Sn chains, we focus on potential electron correlation effects specifically within these chains. Given that the Sn atom has partially filled s or p orbitals in its valence states (with a valence electron configuration of [Kr] $4d^{10}5s^25p^2$), the electron correlation effects from the Sn atom are expected to be negligible. The only potential source of significant electron correlation would be the partially filled d states in the Sc atom (with a valence electron configuration of [Ar] $4s^23d^1$). To investigate this, we have performed additional calculations incorporating a Hubbard U parameter on the Sc $3d$ orbitals (Fig. R11). Our results confirm that the q_3 CDW remains the ground state even with these additional considerations. This consistency across different theoretical approaches further reinforces the reliability of our predictions.

FIG. R11. Total energy of the q_2 and q_3 CDW states relative to the pristine structure as a function of Hubbard U on Sc $3d$ orbital.

R: Lastly, a more challenging yet pertinent question is why this particular kagome material, ScV₆Sn₆, exhibits strong anharmonic effects, whereas other kagome materials do not. Is there a more general principle that can guide the search for other kagome materials with strong anharmonic effects and intriguing CDW or possible superconducting properties?

A: We thank the Reviewer for the interesting comment that leads us to suggest additional CDW materials. ScV₆Sn₆ is not unique in showing strong anharmonic effects, as these are common in materials with CDWs driven by electron-phonon coupling or phonon softening. Well-studied examples include CDWs in various transition metal dichalcogenides, where anharmonic effects have been widely studied [R3–R6]. We also note that, in the prototypical kagome material CsV₃Sb₅, recent theoretical and experimental studies have highlighted strong anharmonic effects [R7, R8]. In that case, while no pronounced phonon softening has been observed experimentally, the anharmonicity is suggested to be strong leading to huge electron-phonon linewidth. This large linewidth could obscure the observation of phonon softening in experiments, even if it is present. We believe that in generic kagome materials exhibiting CDWs, anharmonic effects are likely to be similarly pronounced. Therefore, identifying additional CDW kagome materials would provide intriguing opportunities for exploring anharmonic effects and their related properties. This could ultimately lead to a more general principle applicable to a broader class of CDW materials, offering fruitful directions for future research in this active field.

During revisions, we explored other potential CDW material candidates, leading to the concrete suggestion of ScV₆Pb₆ with a different CDW ground state compared to ScV₆Sn₆. Figure R12a displays the harmonic phonon dispersion of ScV₆Pb₆ (red lines) and ScV₆Ge₆ (black lines). The phonon dispersion of ScV₆Pb₆ shows imaginary modes at the L and H points, indicating CDW instabilities. The calculated total energy (Fig. R12b) confirms that indeed ScV₆Pb₆ has CDW ground states, with the L CDW slightly favoured over the \mathbf{q}_2 CDW. We note that the \mathbf{q}_3 CDW is no longer stable in this compound. In contrast, ScV₆Ge₆ exhibits no imaginary modes, consistent with the calculated total energy showing that the CDW structure is unstable compared to the pristine structure.

Our newly added results reveal an optimal condition for bilayer kagome materials to exhibit CDWs, offering important microscopic mechanism into the CDW instabilities and guiding future experimental studies. Figure R12c shows the bond ratio d_2/d_1 in X-Sc-X chains within the ScV₆X₆ compounds (X=Ge, Sn, and Pb), where the ratio increases as the atomic radius increases from Ge to Pb. A larger bond ratio is one of the necessary conditions for CDWs to emerge in this

FIG. R12. **a** Harmonic phonon dispersions of ScV₆Ge₆ and ScV₆Pb₆. **b** Total energy of various CDW structures relative to the pristine structure in ScV₆X₆ materials (X=Ge, Sn, and Pb). **c** Bond ratio d_2/d_1 in X-Sc-X chains and out-of-plane lattice parameter in the pristine structure of ScV₆X₆ materials (X=Ge, Sn, and Pb).

kagome system, as the CDWs displacement patterns consist of the vibration of X-Sc-X trimers. These vibrations are promoted by the larger bond difference between X-Sc and X-X atoms, which corresponds to a larger bond ratio. In the case of ScV₆Sn₆, the bond ratio is reduced to nearly 1 when Sc is substituted by larger atoms like Y or Gd, leading to the disappearance of CDWs. Similarly, in ScV₆Ge₆, the bond ratio is close to 1, making trimerisation unfavourable and resulting in the absence of CDWs. Additionally, we find that there exists an optimal range of bond ratios that drive CDW formation, as the calculated energy gain of CDW states over the pristine structure diminishes in the Pb case compared to the Sn case, despite Pb having a larger bond ratio.

We note that the \mathbf{q}_3 CDW order is less stable compared to the pristine case in both the Ge and Pb compounds, and thus cannot be observed. In the Pb case, our theory predicts that the L and H (\mathbf{q}_2) CDWs are dominant, which we speculate is due to the larger out-of-plane lattice parameters, where we leave the detailed characterisation as a future work. Our results suggest that partial doping of Pb or Ge at the Sn site could suppress the \mathbf{q}_3 CDW, potentially leading to the dominance of \mathbf{q}_2 CDW. This prediction awaits immediate experimental verification.

Finally, we comment on how strong anharmonic effects could influence other properties such as CDW properties and superconductivity. We expect that the anharmonic effects can be significant in the transport properties of both pristine and CDW phases, as anharmonicity accounts for higher-order phonon-phonon and electron-phonon scatterings. Regarding superconductivity, it

has been shown that anharmonic corrections have significant effects on dynamical stability [R9], and very recently, on superconducting temperatures [R10, R11] in hydrogen based compounds where anharmonicity is strong. Although specific examples demonstrating the importance of anharmonicity are still emerging, the general impact of anharmonicity remains an open question. We expect newly proposed materials to offer valuable opportunities for the exploring the influence of anharmonicity on CDWs, and related superconducting properties in kagome materials.

We have added above figure as new figure 5 and relevant discussions in the revised main text.

R: In summary, while the manuscript provides technically sound results, its findings do not represent a significant advancement over the existing literature and may benefit from being directed to a more specialized journal.

A: We hope that the Reviewer revisits their recommendation after considering our response.

-
- [R1] H. W. S. Arachchige, W. R. Meier, M. Marshall, T. Matsuoka, R. Xue, M. A. McGuire, R. P. Hermann, H. Cao, and D. Mandrus, Phys. Rev. Lett. **129**, 216402 (2022).
- [R2] K. Lejaeghere, G. Bihlmayer, T. Björkman, P. Blaha, S. Blügel, V. Blum, D. Caliste, I. E. Castelli, S. J. Clark, A. D. Corso, S. de Gironcoli, T. Deutsch, J. K. Dewhurst, I. D. Marco, C. Draxl, M. Dułak, O. Eriksson, J. A. Flores-Livas, K. F. Garrity, L. Genovese, P. Giannozzi, M. Giantomassi, S. Goedecker, X. Gonze, O. Grånäs, E. K. U. Gross, A. Gulans, F. Gygi, D. R. Hamann, P. J. Hasnip, N. A. W. Holzwarth, D. Iușan, D. B. Jochym, F. Jollet, D. Jones, G. Kresse, K. Koepnik, E. Küçükbenli, Y. O. Kvashnin, I. L. M. Locht, S. Lubeck, M. Marsman, N. Marzari, U. Nitzsche, L. Nordström, T. Ozaki, L. Paulatto, C. J. Pickard, W. Poelmans, M. I. J. Probert, K. Refson, M. Richter, G.-M. Rignanese, S. Saha, M. Scheffler, M. Schlipf, K. Schwarz, S. Sharma, F. Tavazza, P. Thunström, A. Tkatchenko, M. Torrent, D. Vanderbilt, M. J. van Setten, V. V. Speybroeck, J. M. Wills, J. R. Yates, G.-X. Zhang, and S. Cottenier, Science **351**, aad3000 (2016).
- [R3] R. Bianco, L. Monacelli, M. Calandra, F. Mauri, and I. Errea, Phys. Rev. Lett. **125**, 106101 (2020).
- [R4] J. S. Zhou, L. Monacelli, R. Bianco, I. Errea, F. Mauri, and M. Calandra, Nano Lett. **20**, 4809 (2020).
- [R5] J. Diego, A. H. Said, S. K. Mahatha, R. Bianco, L. Monacelli, M. Calandra, F. Mauri, K. Rossnagel, I. Errea, and S. Blanco-Canosa, Nat. Commun. **12**, 598 (2021).
- [R6] A. O. Fumega, J. Diego, V. Pardo, S. Blanco-Canosa, and I. Errea, Nano Lett. **23**, 1794 (2023).
- [R7] G. He, L. Peis, E. F. Cuddy, Z. Zhao, D. Li, Y. Zhang, R. Stumberger, B. Moritz, H. Yang, H. Gao, T. P. Devereaux, and R. Hackl, Nat. Commun. **15**, 1895 (2024).
- [R8] M. Gutierrez-Amigo, . Dangić, C. Guo, C. Felser, P. J. Moll, M. G. Vergniory, and I. Errea, arXiv preprint arXiv:2311.14112 (2023).
- [R9] I. Errea, F. Belli, L. Monacelli, A. Sanna, T. Koretsune, T. Tadano, R. Bianco, M. Calandra, R. Arita, F. Mauri, and J. A. Flores-Livas, Nature **578**, 66 (2020).
- [R10] F. Belli and I. Errea, Phys. Rev. B **106**, 134509 (2022).
- [R11] Dangić, L. Monacelli, R. Bianco, F. Mauri, and I. Errea, Commun. Phys. **7**, 150 (2024).

Response to Reviewer 1

Reviewer 1 (R) : The authors have satisfactorily addressed my previous comments. The quality of the revised manuscript seems to be much enhanced, and the readability is much improved.

Authors (A): We thank the Reviewer for reviewing our revised manuscript. We appreciate the reviewer's positive feedback on the improvements made to the manuscript.

R: There is just one more point I would like to raise. To be fair, the strong electron-phonon coupling effect and short-range order ($1/3, 1/3, 1/2$) was also discussed (perhaps the first) in Ref. 39. So I would suggest the authors to add Ref. 39 to the appropriate positions in the introductions.

A: We thank the Reviewer for raising this point. We have now added this reference to the relevant positions in the introduction to acknowledge the early discussion of strong electron-phonon coupling effects and the short-range order.

Response to Reviewer 2

Reviewer 2 (R) : The authors have addressed all issues raised in my previous report. I now recommend its publication in Nature Communications.

Authors (A): We thank the Reviewer for reviewing our revised manuscript. We are happy to receive the recommendation of accepting our manuscript.

Response to Reviewer 3

Reviewer 3 (R) : In the revised manuscript, the authors have addressed the majority of my previous comments and made improvements to the overall clarity and depth of the study. The newly added discussion on Ge and Pb substitution for Sn is a valuable addition that could potentially inspire further experimental investigations in bilayer kagome systems. I am inclined to recommend the manuscript for publication, contingent upon addressing the following minor points.

A: We are grateful to the Reviewer for taking the time and effort to read our revised manuscript. We also thank the reviewer for the positive feedback and for recognizing the improvements made in the revised manuscript.

R: 1. While the authors have provided an argument regarding the choice of semi-core pseudopotentials, I remain somewhat unconvinced. Although the DFT calculations are technically sound and are supported by full-electron calculations using WIEN2K, the observed energy difference is still as small as 1 meV. This raises the possibility that other factors, which may not have been fully accounted for in the present DFT framework, could be influencing the results. That said, given that self-consistent anharmonic calculations and full-electron WIEN2K calculations are regarded as the most accurate DFT method, it is understandable that there may be limitations to what can be further addressed here. Nevertheless, I suggest that the authors include a brief discussion on this small energy difference in the manuscript to provide context.

A: We understand the reviewer's concern regarding the small energy difference, but we would like to emphasize that relative energies and trends (e.g. energy as a function of lattice parameter) tend to be more robust than absolute energies. And indeed, as the Reviewer recognises, our model represents the state-of-the-art of what is possible computationally. In addition, our proposed new regimes with strain and doping can be verified in future experimental studies, which would further confirm the validity of our model. We also note that the small energy difference between the two CDWs is an intrinsic feature of this kagome system, highlighting the competing nature of the two CDWs. As suggested by the Reviewer, we have added a brief discussion on the small energy difference in the revised manuscript as

"...Overall, this demonstrates that the inclusion of semi-core electron states in the valence is

necessary to obtain a theoretical model that correctly predicts the ground state CDW order of ScV_6Sn_6 . This explains and resolves the outstanding discrepancy between theory and experiment, and provides a foundation for the predictive model of the CDW state in bilayer kagome ScV_6Sn_6 described above. **Additionally, we note that the calculated energy difference between the two CDWs is small (less than 2 meV/f.u.), highlighting the competing nature of the two CDWs in ScV_6Sn_6 . This suggests that the competition between the two CDW orders can be easily manipulated via external perturbations, as discussed in detail below...."**

R: 2. Regarding the density of states for the q_2 CDW phase with semi-core states, there is a small peak at the Fermi level associated with contributions from Sc and Sn1 atoms. This feature is somewhat unexpected, as one would typically anticipate that a CDW phase would reduce the DOS at the Fermi level, rather than produce a peak. Could the authors elaborate on the origin of this peak? Specifically, does this feature persist in the DOS without the presence of the CDW? Additionally, could this peak be related to the total energy difference between the q_2 and q_3 CDW phases, given that the q_3 CDW does not exhibit such peaks?

A: We appreciate this comment, which led us to correct an error in the Fermi level energy used when plotting the electronic structures (shifted by 30 meV; all other properties remain unchanged apart from this Fermi level shift). In the revised manuscript, we have updated the Fermi level energy of the q_2 CDW state as shown in Fig. R1 (and have correspondingly revised Fig. 3 in the main text and Supplementary Figure S7).

FIG. R1. Electronic band structures and density of states (DOS) calculated with and without treating semi-core states as valence states for the q_2 CDW state.

This correction shifts the unexpected small peak above the Fermi level. When compared with the DOS of the pristine structure (Fig. R2), both \mathbf{q}_2 and \mathbf{q}_3 CDW states show a depletion of DOS exactly at the Fermi level, consistent with the expected behaviour that a CDW phase would reduce the DOS at the Fermi level, as pointed out by the Reviewer. Further inspection reveals a notable difference in the atom-projected DOS, particularly for the Sc and Sn1 states, which accounts for the energy difference between the two CDW states.

FIG. R2. Electronic band structures and density of states (DOS) calculated with and without treating semi-core states as valence states for the \mathbf{q}_2 CDW state.

R: If these minor comments can be addressed, I believe the manuscript would be suitable for publication.

A: We thank the Reviewer for their helpful comments and suggestions. We have carefully addressed each of the minor comments in this revision and hope that our responses and changes have made the manuscript suitable for publication.